# Reduced IFN-ß inhibitory activity of Lagos bat virus phosphoproteins in human compared to Eidolon helvum bat cells

Jan Papies[1], Andrea Sieberg[1], Daniel Ritz[2], Daniela Niemeyer[1,3], Christian Drosten[1,3], Marcel A. Müller[1,3,4]*

1 Institute of Virology, Charité-Universitätsmedizin Berlin, Corporate Member of Freie Universität Berlin and Humboldt-Universität zu Berlin, Berlin, Germany, 2 Institute of Virology, Universitätsklinikum Bonn, Bonn, Germany, 3 German Centre for Infection Research (DZIF), Partner Site Berlin, Berlin, Germany, 4 Martsinovsky Institute of Medical Parasitology, Tropical and Vector Borne Diseases, Sechenov University, Moscow, Russia

* marcel.mueller@charite.de

**Data Availability Statement:** All relevant data are within the manuscript and its Supporting Information files.

## Abstract

*Eidolon helvum* bats are reservoir hosts for highly pathogenic lyssaviruses often showing limited disease upon natural infection. An enhanced antiviral interferon (IFN) response combined with reduced inflammation might be linked to the apparent virus tolerance in bats. Lyssavirus phosphoproteins inhibit the IFN response with virus strain-specific efficiency. To date, little is known regarding the lyssavirus P-dependent anti-IFN countermeasures in bats, mainly due to a lack of *in vitro* tools. By using *E. helvum* bat cell cultures in a newly established bat-specific IFN-promoter activation assay, we analyzed the IFN-ß inhibitory activity of multiple lyssavirus P in *E. helvum* compared to human cells. Initial virus infection studies with a recently isolated *E. helvum*-borne Lagos bat virus street strain from Ghana showed enhanced LBV propagation in an *E. helvum* lung cell line compared to human A549 lung cells at later time points suggesting effective viral countermeasures against cellular defense mechanisms. A direct comparison of the IFN-ß inhibitory activity of the LBV-GH P protein with other lyssavirus P proteins showed that LBV-GH P and RVP both strongly inhibited the bat IFN-β promotor activation (range 75–90%) in EidLu/20.2 and an *E. helvum* kidney cell line. Conversely, LBV-GH P blocked the activation of the human IFN-β promoter less efficiently compared to a prototypic Rabies virus P protein (range LBV P 52–68% vs RVP 71–95%) in two different human cell lines (HEK-293T, A549). The same pattern was seen for two prototypic LBV P variants suggesting an overall reduced LBV P IFN-ß inhibitory activity in human cells as compared to *E. helvum* bat cells. Increased IFN-ß inhibition by lyssavirus P in reservoir host cells might be a result of host-specific adaptation processes towards an enhanced IFN response in bat cells.

## Introduction

Lyssaviruses infect a wide range of mammals causing up to 50,000 fatal rabies cases in humans per year [1]. Most lyssaviruses are found in bats and transmissions to humans are sporadically

**Funding:** This study was supported by the German Research Foundation (DFG grant DR 772/10-2 to C.D.; https://www.dfg.de/) and the Volkswagen Foundation (AZ93345 to M.A.M.; https://www.volkswagenstiftung.de/). The funders had no role in study design, data collection and analysis, decision to publish, or preparation of the manuscript.

**Competing interests:** The authors have declared that no competing interests exist.

reported [2–6]. *Eidolon helvum* (*E. helvum*) is Africa's most widespread fruit bat species and the putative natural reservoir host for Lagos bat virus (LBV), an African lyssavirus with sporadic transmission to other mammals [7–10]. Upon virus entry, initial replication can trigger the antiviral interferon (IFN) pathway of hosts' innate immune system (IFN induction), which results in the expression of IFN-stimulated antiviral genes (IFN signaling) [11]. To limit the expression of genes with antiviral functions, lyssaviruses encode a multifunctional P protein. The well-studied Rabies virus (RABV) P protein (RVP) blocks several steps of the IFN pathway at the level of IFN induction and signaling, modulating pathogenicity *in vitro* and *in vivo* [12–15]. During IFN induction, RVP interferes with TANK-binding kinase (TBK)- and inhibitor of nuclear factor kappaB kinase (IKK)-mediated activation of IFN regulatory factor 3 (IRF-3) and blocks its nuclear translocation, thereby preventing transcription of IFN genes [16]. In addition, RVP also interacts with STAT proteins downstream of IFN receptor activation to block IFN signaling and IFN-stimulated gene (ISG) transcription [17]. In contrast to the P protein of RABV, there are fewer studies on LBV P proteins available, and all were performed in human cells. Whereas some street strain lyssavirus-derived P proteins seem to have enhanced efficiency to block IFN induction in human cells [18], inhibition of IFN signaling by P proteins was shown to be highly conserved among lyssaviruses, including LBV [19]. Although IFN antagonism is well described in terrestrial mammals, the interplay with bat immunity remains vague, particularly for bat-associated lyssaviruses.

Bats harbor many highly pathogenic viruses [20–24] including close relatives of the currently circulating SARS-CoV-2 [25, 26] underscoring the importance of bats as viral reservoirs [27, 28]. While most of these viruses do not cause disease in bats, lyssavirus infection can result in clinical disease manifestation [29–31]. A recent *in vivo* study showed that *E. helvum* bats develop severe disease upon intracranial inoculation with LBV [10]. However, studies using more natural routes of infection reported limited lyssavirus infectivity and mortality in several bat species [12, 32]. High anti-LBV seroprevalence was detected in wild *E. helvum* colonies, suggesting the occurrence of mild or asymptomatic infections [7, 8, 10]. While these findings support the idea of intrinsic resistance of bats towards lyssavirus infection, the underlying immunological mechanisms remain unknown.

Current research indicates reduced inflammatory responses towards viral infection in bats while maintaining profound IFN-related antiviral actions [25, 33, 34]. We previously analyzed virus-infected bat cell lines and combined *in vitro* virus replication data with modeling approaches towards virus-host interaction [35]. The identified enhanced IFN-mediated antiviral defense in bat cells suggested a putative IFN-dependent selection pressure on bat-borne viruses. Consequently, bat-adapted viruses would evolve highly efficient IFN antagonists to allow virus replication in this IFN-rich environment. Virus adaptation might thus promote intra-host transmission without severe damage to host bat cells but possibly result in high virulence in novel hosts [35]. These initial findings encourage more in-depth functional analyses to elucidate the complex interactions of bat viruses in their natural host.

Studying bat-borne viruses using bat *in vivo* models is costly and difficult, in particular, in the case of biosafety level 3 and 4 viruses [36]. Rabies virus and related lyssaviruses were shown to infect epithelial cells both *in vitro* and *in vivo*, including lung and kidney cells of bats [37–40]. Regarding *in vitro* studies, most experiments involving bat-associated viruses like LBV are performed in primate or rodent cells [18, 19]. It is unclear if characteristics from viruses of terrestrial mammals can readily be applied to bat-associated viruses. Furthermore, the distinctive features of bat immunity must be considered when analyzing endemic bat viruses. Therefore, the development of suitable bat cell cultures and specific immune assays that allow targeted analyses presents a feasible alternative for studying bat viruses in the reservoir host context.

Here we developed bat-specific assays to compare distinct features of lyssavirus P proteins in cells of a natural lyssavirus host, *E. helvum*. We found conserved and strong IFN-ß inhibitory activity of all tested lyssavirus P proteins in bat cells. In human cells, the activity of several different LBV P was markedly reduced compared to RVP. Our data encourage further studies of the complex interplay of bat-associated viruses and host cells in the natural reservoir context.

## Material and methods

### Ethics statement

Permission to capture and sample bats was obtained from the Wildlife Division, Forestry Commission, Accra, Ghana. Bats were captured at a sampling site in Kumasi, Ghana with following geographic coordinates: N06˚42′02.0″ W001˚37′29.9″. Animal handling and all involved procedures were performed by trained veterinary staff and all efforts were made to minimize the suffering of animals. Bats were caught using mist nets and animals were anesthetized with a Ketamine/Xylazine mixture and euthanized by cervical dislocation as covered by the obtained permits (permit no. CHRPE49/09; A04957), as described previously [41, 42]. Samples were exported under a state contract between the Republic of Ghana and the Federal Republic of Germany under export permission (permit no. CHRPE49/09; A04957) from the Veterinary Services of the Ghana Ministry of Food and Agriculture.

### Cells and culture conditions

HEK-293T (*Homo sapiens*, kidney; ATCC CRL-3216), A549 cells (*Homo sapiens*, lung; ATCC CCL-185), VeroFM (*Chlorocebus sp.*, kidney; ATCC CCL-81), and EidNi/41.3 (*E. helvum*, kidney; [42]) were grown in Dulbecco's Modified Eagles Medium (DMEM, Gibco) supplemented with 1% Penicillin/Streptomycin (Gibco), 1% non-essential amino acids (Gibco), 1% sodium pyruvate (Gibco), 1% L-glutamine (Gibco) and 10% FBS (Gibco) at 37˚C and 5% $CO_2$.

For the generation of the *E. helvum* lung cell culture EidLu/20.2, bats were caught with mist nets under the supervision of Ghana authorities. Animals were anesthetized with a Ketamine/Xylazine mixture and euthanized to perform organ preparations (permit no. CHRPE49/09; A04957) as described previously [42]. Cells were immortalized by transduction of the large T antigen of Simian virus 40 (SV40). Subsequently, cells were expanded, subcloned, and frozen at -80˚C for cryopreservation. EidLu/20.2 and EidNi/41.3 cells were genotyped by cytochrome b PCR as previously described [43, 44]. All cells were tested negative for contaminations with mycoplasma [45], simian virus 5 (in-house assay), lyssaviruses [46], and filoviruses [47] by RT-PCR.

### Generation of VSV-RNA

VeroFM cells ($3x10^7$) were infected with vesicular stomatitis virus (VSV) diluted in DMEM at a multiplicity of infection (MOI) of 0.01 plaque-forming units/ml. After 18 hours (h) cells were washed with phosphate-buffered saline (PBS) and lysed in 2.5 ml of β-mercaptoethanol-containing buffer RA1 from the NucleoSpin RNA kit (Macherey-Nagel). The lysate was incubated at -80˚C overnight and RNA extraction was performed according to the manufacturer´s instructions.

### Transfection of eukaryotic cells

Transient transfection of eukaryotic cells was performed using FuGENE® HD (Promega) and X-tremeGENE™ siRNA (Roche) according to the manufacturers´ instructions. Briefly, cells

$(1x10^5$ to $4x10^5$ cells/ml) were cultured in 24-well plates and plasmid DNA was added after 15 minutes (min) incubation with transfection reagent in OptiPRO SFM™ (Gibco) at a 1:3 ratio. Transfection of total RNA from VSV-infected cells was performed with X-tremeGENE™ siRNA analogous to FuGENE® HD transfection.

### *In vitro* growth kinetics with LBV

A549 and EidLu/20.2 cells were infected with low-passage LBV Ghana isolate (LBV-GH; passage 4; INSDC LN849915) for comparative virus replication kinetics. Cells were seeded and incubated for 24 h and infected with LBV-GH with an MOI of 0.001 in OptiPRO SFM (Gibco). After a 1 h incubation at 37˚C, the inoculum was removed, cells were washed once with PBS and supplemented with DMEM. Supernatants were collected at the indicated time points (18, 24, 48, 72, 96 h) and 75 µl were subjected to viral RNA isolation with the NucleoSpin® RNA virus kit (Macherey-Nagel) according to the manufacturer´s instructions. Subsequently, viral replication was monitored by genotype-specific LBV RT-qPCR using the Superscript III One-step RT-PCR kit (Thermo Fisher) according to the following protocol: reactions were performed in 12.5 µl volume with 6.25 µl of 2x reaction buffer, 10 µM forward and reverse primers (see S1 Table for details), 5 µM probe, 0.5 µl of bovine serum albumin (BSA), 1.5 µl of RNase-free water, 0.5 µl of enzyme mix, and 100 ng of RNA template. RNA was reverse-transcribed at 55˚C for 20 min, followed by an initial denaturation at 95˚C for 120 seconds (s). Cycling and fluorescence signal acquisition was done for 45 cycles of 95˚C for 15 s and 58˚C for 40 s using the LightCycler® 480 Real-Time PCR System (Roche). Additionally, previously generated LBV *in vitro* transcribed RNA standards (in-house, unpublished) were used for absolute quantification of viral RNA.

### IFN bioassay

To investigate the functionality of IFN signaling pathways in the analyzed cells IFN bioassays were performed as described before [48]. Briefly, cells were cultured in 24-well plates for 24 h and the IFN response was induced by transfection of 0.5 µg, 0.75 µg, or 1.5 µg VSV-RNA with X-tremeGENE™ siRNA transfection reagent according to the manufacturer's protocol. After 20 h, supernatants were collected and serially diluted in used DMEM from the respective cultured cell line. Next, naïve cells were incubated with 100 µl of the diluted supernatants for 7 h, followed by infection with a Rift Valley fever virus *Renilla* luciferase-based reporter virus (RVFV-Ren) [48] at an MOI of 0.002 for 16 h. Cell lysis and read-out were performed with the *Renilla* Luciferase Assay System (Promega) according to an adapted protocol. Briefly, 16 h post-infection (hpi) supernatants were removed and cells were lysed in 50 µl *Renilla* luciferase assay lysis buffer (Promega). Luminescence was measured using the Synergy™ 2 microplate reader (BioTek).

Additionally, cells were treated with serial dilutions of recombinant universal type I IFN (rIFN; R&D Systems) ranging from 1 Unit (U)/ml to 1,000 U/ml IFN and infected with RVFV-Ren. Cell lysis and analysis were performed as described previously.

### Nucleic acid extraction and RT-qPCR assays

Isolation of total RNA was performed using the NucleoSpin® RNA kit according to the manufacturer´s instructions. Nucleic acid concentration was determined by photometric analysis with the NanoDrop 2000c (Peqlab).

To quantify *IFNB1* RNA levels a hydrolysis probe-based reverse transcription quantitative PCR (RT-qPCR) assay was used. Primers and probes for *IFNB1* and *ACTB* for *H. sapiens* and *E. helvum* were designed as described previously [42] and purchased from Integrated DNA

Technologies. RT-qPCRs were performed using the Superscript III One-step RT-PCR kit (Thermo Fisher) with primers and probes according to S1 Table. Cycling and fluorescence signal acquisition was done for 45 cycles of 95˚C for 15 s and 58˚C for 40 s. RT-qPCR and data processing was performed using the LightCycler® 480 Real-Time PCR System (Roche). Relative quantification of *IFNB1* against the reference gene *ACTB* was evaluated using the $2^{-\Delta\Delta Ct}$ method as described elsewhere [49]. Quantification of *CCL5*, *IFIT1*, and *MX1* expression was performed in an analogous procedure, using *TBP* as reference gene. The primers and probes that were used in this study are listed in S1 Table.

## Directional cloning

Generation of recombinant pCAGGS expression vectors and pGL4 reporter plasmids was performed using directional cloning with restriction enzymes KpnI and NotI. Restriction recognition sites were inserted into DNA sequences by PCR. Subsequently, vector and insert DNA were digested and ligated using the Rapid DNA Ligation kit (Roche) with 10 ng of digested vector and 30 ng of insert DNA. Next, One Shot® TOP10 chemically competent *E. coli* cells (Thermo Fisher) were transformed with recombinant vector DNA according to the manufacturer´s instructions. Clones were checked for recombinant DNA by colony PCR using the indicated primers. Plasmid DNA was isolated using the NucleoSpin® Plasmid kit and the NucleoBond® Xtra Midi EF kit (Macherey-Nagel) according to the manufacturer's instructions. Sequence verification of recombinant DNA was accomplished by Sanger sequencing (Microsynth AG) and analysis was performed with Geneious version 9.1.8 software (Biomatters Limited).

## Generation of expression plasmids

Plasmids encoding lyssavirus P proteins were generated by directional cloning of the respective lyssavirus P coding sequence (CDS) into the pCAGGS expression vector (sequence IDs of applied lyssavirus species/strains and phosphoproteins: RVP: EU886634.1; RVP 1088: AB645847.1; DUVV P: AF049115.1; LBV GH P: INSDC LN849915; LBV Sen P: NC_020807.1; LBV Nig P: EF547407.1). Constructs were designed containing restriction enzyme recognition sites, N-terminal FLAG-tag, and a Kozak sequence. DNA was synthesized as gBlocks® Gene Fragments by Integrated DNA Technologies. Cloning into pCAGGS vector and DNA sequencing was accomplished as described above.

## Gene expression analysis by Western blot and immunofluorescence assay

Cells were grown in 24-well plates and transfected with 0.5 μg of the above-mentioned expression plasmids. For Western blot analysis cell lysates of the P protein-expressing cells were prepared. Cells were washed with cold PBS and lysed with Pierce RIPA lysis buffer (Thermo Fisher), supplemented with Benzonase® (Sigma-Aldrich), dithiothreitol (DTT), and Proteinase Inhibitor Cocktail III (Sigma-Aldrich). After incubation on ice for 20 min, cell lysates were centrifuged and stored at -80˚C until analysis.

Analysis of protein lysates was performed as described elsewhere [50]. Briefly, protein lysates along with the Spectra™ Multicolor Broad Range Protein Ladder (Thermo Fisher) were separated using 12% polyacrylamide gels (120 V, 45 min). Western blot analysis was performed with monoclonal mouse-anti-FLAG (1:5,000; Sigma-Aldrich; F1804) and monoclonal pan-species mouse-anti-β-actin (1:5,000; Sigma-Aldrich; A5316) primary antibodies that detect conserved epitopes of bat β-actin. For signal detection, horseradish peroxidase (HRP) labeled polyclonal goat-anti-mouse secondary antibody (1:10,000; Dianova; 115-035-146) was used. Chemiluminescence signal was detected using the SuperSignal® West Femto

Chemiluminescence Substrate (Thermo Fisher) and visualized with the Fusion FX7™ imaging system (Vilber Lourmat).

For indirect immunofluorescence assay (IFA) analysis cells were seeded onto glass cover-slips in 24-well plates and transfected with pCAGGS and pCAGGS-P carrying expression plasmids. Cells were fixed with Histofix® (Carl Roth), permeabilized with Triton™ X-100 (Sigma Aldrich) in PBS and incubated with a mouse-anti-FLAG (1:100; Sigma-Aldrich) primary antibody. Secondary detection was performed with a Cy3-conjugated goat-anti-mouse (1:200; Dianova) antibody. Glass slides were mounted using ProLong® Gold Antifade Reagent with DAPI (Thermo Fisher) and analyzed by fluorescence microscopy.

## Sequencing of unknown bat IFN-β promoter sequences

IFN-β promoter sequences of *E. helvum* [42], *Hypsignathus monstrosus* [51], *Rhinolophus alcyone* [52], *Tadarida brasiliensis* (ATCC CCL-88), and *Myotis daubentonii* [51] cells were amplified by PCR and sequenced using primers Eidolon IFNb-P Screen fw, Eidolon IFNb-P Screen rev MyoLucF378, MyoLucR1082, bIFNp F1, bIFNpF2, bIFNp R3.Afr, according to S1 Table. Sequence verification was accomplished by Sanger sequencing (Microsynth AG) and analysis was performed with Geneious version 9.1.8 software (Biomatters Limited). GenBank entries were uploaded for the respective sequences of *E. helvum* (accession no. MK762750); *Hypsignathus monstrosus* (*H. monstrosus*, accession no. MK762749); *Rhinolophus landeri* (*R. landeri*, accession no. MK762752); *Tadarida brasiliensis* (*T. brasiliensis*, accession no. MK762751); and *Myotis daubentonii* (*M. daubentonii*, accession no. MK762753).

## Establishment of a bat-specific IFN-β promoter activation reporter assay

Recombinant IFN-β reporter vectors were produced by directional cloning of human and *Rousettus aegyptiacus* (*R. aegyptiacus*) IFN-β promoter sequences into the pGL4.10[luc2] vector (Promega), encoding a genetically engineered firefly luciferase. Restriction recognition sites were introduced to the promoter sequence via PCR using 20 ng of previously generated plasmids containing the human IFN-β promoter sequence (p125Luc; Prof. Takashi Fujita, University of Kyoto) and the in-house amplified *R. aegyptiacus* IFN-β promoter sequence, respectively, with primers p189_KpnI_fw2, p189_complete_XhoI_rev2, and p125_KpnI_fw2 and p125_complete_XhoI_rev2 according to S1 Table. The cloning procedure for vector and IFN-β promoter sequences was performed as described above. Clones were checked for recombinant DNA by colony PCR using primers pGL4.10-luc2_MCS_Screen_fw and pGL4.10-luc2_MCS_Screen_rev (see S1 Table for details). DNA sequence verification was performed as described above.

## Dual-luciferase promoter activation reporter assays

Activation of the human IFN-β promoter was analyzed by IFN-β promoter activation reporter assay. HEK-293T and A549 cells were seeded in 24-well plates ($1 \times 10^5$ cells/ml) and transfected with 5 ng pGL4.74[hRluc/TK] plasmid (Promega), 200 ng of the pGL4.10[*luc2*] plasmid (Promega) including the human IFN-β promoter sequence, and 50 ng of either empty pCAGGS vector or the indicated pCAGGS expression plasmids. The IFN-β promoter activation reporter assay in EidLu/20.2 and EidNi/41.3 bat cells was performed accordingly with the following adaptations: bat cells were transfected with 25 ng pRL-SV40 plasmid (Promega) and 250 ng pGL4.10[luc2] plasmid including the *R. aegyptiacus* IFN-β promoter. After 24 h the cells were transfected with 1 µg of total RNA from vesicular stomatitis virus (VSV) infected cells using X-tremeGENE™ siRNA transfection reagent. After 16 h cells were lysed with 100 µl of passive lysis buffer (Promega) and luciferase luminescence was measured with the Dual-Luciferase®

Reporter Assay System (Promega) using the Mithras LB 940 multimode microplate reader (Berthold Technologies). IFN-β promoter activity was calculated as percent induction relative to the empty vector control. Intra-assay variations in transcriptional activity were compensated by normalization with pGL4.74[hRluc/TK]-based luminescence data. Statistical analysis was performed using independent t-tests and SOFA Statistics version 1.4.6. Optimization experiments were performed accordingly and as indicated in the text and figure legends.

Essentially, the ISG54/interferon-stimulated response element (ISRE) promoter activation reporter assay was performed in an analogous procedure. Briefly, the following variations were applied to the protocol: HEK-293T cells were transfected with 25 ng pRL-SV40 and 125 ng pISG54-luc [53] reporter plasmids, and promoter activation was achieved by incubating cells with 200 U/ml rIFN for 18 h before cell lysis.

## Results

### Interferon-competent EidLu/20.2 show enhanced LBV replication compared to A549 cells

To conduct *in vitro* studies in reservoir-derived cells, we established an immortalized monoclonal *E. helvum* bat lung cell line, designated EidLu/20.2 (Fig 1A) from a previously published heterogeneous EidLu cell culture [54]. To confirm that immortalization and subcloning did not affect the integrity of the IFN pathway, we compared the IFN response of EidLu/20.2 with the prototypic human lung cell line A549. First, we addressed IFN-β mRNA expression levels. We triggered the IFN response according to our previous experience with an RVFV-Ren reporter virus that lacks its IFN antagonist NSs and is predominantly sensed by RIG-I [42, 48, 55]. In addition, we transfected total RNA from VSV-infected cells previously shown to trigger both RNA helicases RIG-I and MDA5 [42]. The IFN-β mRNA induction ranged from 1,500-fold in VSV-RNA-transfected A549 cells to a maximal 400,000-fold in RVFV-Ren-infected EidLu/20.2 cells. Overall, we found a 10-fold higher IFN-β mRNA expression in EidLu/20.2 cells compared to A549 for both stimuli (Fig 1B). Secondly, we analyzed the secretion of bioactive IFN and characterized the cell-specific IFN sensitivity. We applied a previously established species-independent IFN bioassay [42, 49, 56, 57]. Cells were pretreated with supernatants from VSV-RNA-transfected or untreated control cells for 8 h and subsequently

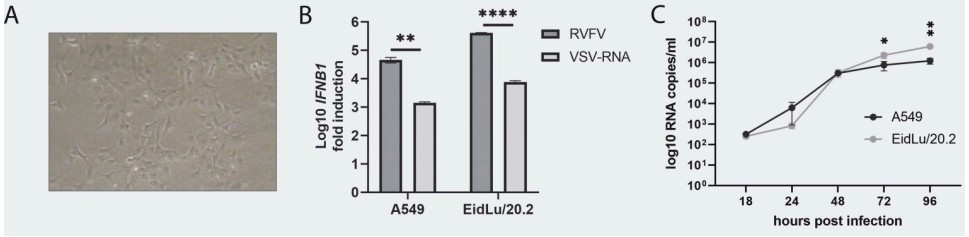

**Fig 1. LBV infection of IFN-competent human and bat cells.** (A) Primary cell cultures were generated from *E. helvum* lung tissue, immortalized by lentiviral transduction of the simian virus 40 large T antigen, and subcloned by endpoint dilution, generating a clonal cell line designated EidLu/20.2. (B) The IFN response was induced using either infection with a Rift Valley fever virus-*Renilla* reporter virus (RVFV-Ren) (MOI 0.2) or transfection of 0.75 μg VSV-RNA per 24-well. 8 h post-treatment RNA was extracted and subjected to RT-qPCR analysis of *IFNB1* expression. Relative quantification was performed using the $2^{-\Delta\Delta Ct}$ method and actin-β (*ACTB*) as a reference gene; induction is presented as Log10 fold induction. (C) A549 and EidLu/20.2 cells were infected with LBV-Ghana isolate (MOI 0.001). Viral RNA was extracted from cell culture supernatants and RT-qPCR was performed. Absolute quantification was done using LBV RNA standard curves. Data are presented as mean values and SD, derived from three biological replicates. All experiments were performed at least in triplicate. **** = p<0.0001; ** = p<0.01; * = p<0.05; ns = not significant; determined by student's t-test (see S2 Table for details).

infected with the IFN-sensitive RVFV-Ren reporter virus. High levels of bioactive IFN cell supernatants correspond to delayed recovery of RVFV-Ren growth in diluted supernatants (shown as relative light units (RLU)) [48]. Whereas preincubation of cells with a 1:100 dilution of A549-derived supernatants resulted in only 40% RVFV growth, EidLu/20.2 already showed 80% RVFV growth recovery (S1A and S1B Fig; experimental procedure is illustrated in S1C Fig) suggesting reduced IFN secretion by EidLu/20.2 cells or altered IFN signaling dynamics. Based on these findings, we sought to further investigate the interplay between bat-associated viruses and the host IFN response. To initially compare LBV replication between human and bat cells we infected A549 and EidLu/20.2 with a low-passage bat-borne LBV isolate from *E. helvum* brain named LBV-GH [8] using an MOI of 0.001. Virus growth was assessed in cell supernatants by genotype-specific RT-qPCR over a time course of 96 h (Fig 1C). LBV-GH replication was comparable between A549 and EidLu/20.2 cells at early time points (18 to 48 h) reaching up to $3.3 \times 10^5$ RNA copies/ml in EidLu/20.2 cells (A549: $3 \times 10^5$ RNA copies/ml; Fig 1C). Final virus titers were significantly higher in EidLu/20.2 cells at 96 hpi reaching $6.2 \times 10^5$ RNA copies/ml as compared to $1.2 \times 10^5$ copies/ml in A549 cells. The higher final titers of LBV-GH in EidLu/20.2 might be linked to the observed reduction of IFN bioactivity in this cell line. Another factor possibly affecting LBV propagation and RNA levels in human and bat cells might be virus-dependent control of the cellular antiviral response.

### IFN-β promoter activation by LBV P is reduced in human cells

Cellular countermeasures against lyssavirus infections might modulate virus replication and often involve the IFN response [14, 16, 19]. The highly conserved lyssavirus phosphoprotein (P protein) counteracts antiviral cellular responses of innate immunity [16, 19, 57]. To first evaluate the impact of LBV P on IFN induction and signaling in human cells we cloned the P protein of the LBV-GH isolate and performed established promoter activation reporter assays for IFN-β and ISG54/ISRE signaling [53] in human HEK-293T cells. As control and for comparison, we included RVP, known to efficiently block IFN induction and signaling in human cells [16, 19, 57]. As expected, RVP reduced the IFN-β (Fig 2A) and ISG54/ISRE (Fig 2B) promoter activation by more than 90%. Importantly, we detected a significantly reduced inhibitory effect on IFN-β promoter activation by LBV P (63%) as compared to RVP (95%) (Fig 2A). At the same time, LBV P and RVP exhibited a comparable reduction of IFN-based ISG54/ISRE promoter activation (LBV P and RVP both >90%), with LBV P showing slightly increased efficacy (Fig 2B). To validate whether the increased activity of LBV P over RVP in blocking ISRE activation translates into meaningful differences in the expression of distinct downstream interferon-stimulated genes (ISG), we quantified ISG transcription in RVP or LBV P expressing HEK-293T cells (Fig 2C). As expected, we observed a robust and highly comparable inhibition of *CCL5*, *IFIT1*, and *MX1* induction by RVP and LBV P in IFN-treated cells. In contrast, there was no significant difference detectable between RVP and LBV P efficacy to block ISRE activation. In addition, we assessed the activation of the conserved ISG54/ISRE reporter in EidLu/20.2 bat cells (S1D Fig). We detected robust ISRE activation upon recombinant IFN treatment and also observed a strong inhibition of ISG54/ISRE activation by RVP and even stronger for LBV P. As the pattern of IFN signaling inhibition by RVP and LBV P was comparable between bat and human cells we focused our study on the inhibitory activity of lyssavirus P proteins on IFN-β induction.

### Establishment of a bat-specific IFN-β promoter activation reporter assay

The differential inhibition of IFN-β promoter activation by RVP and LBV P in human cells and the fact that functional aspects of lyssavirus P proteins have mainly been investigated in

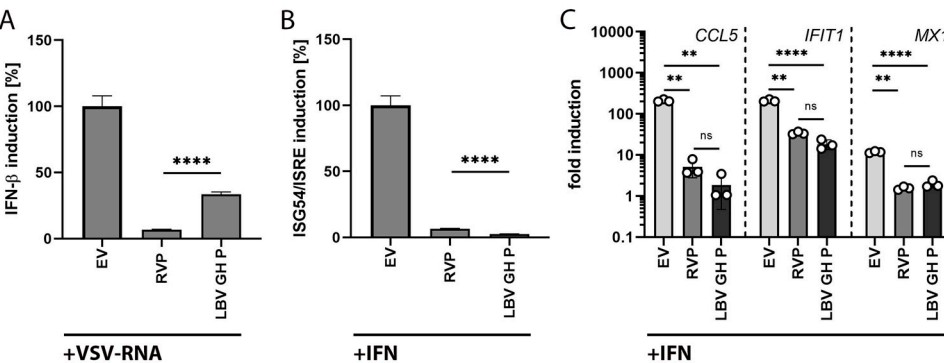

**Fig 2. Impact of lyssavirus phosphoproteins on IFN-β and ISG54/ISRE promoter activation and ISG expression.**
Human HEK-293T cells were transfected with RL and FF luciferase reporter plasmids (A: pGL4.10[*luc2*]; B:
pISG54-luc) and 50 ng of pCAGGS empty vector (EV) or pCAGGS vector encoding Rabies virus P protein (RVP) or
LBV-GH P protein (LBV GH P). Cells were stimulated by VSV-RNA transfection (A) or recombinant IFN treatment
(B) and lysed after 18 h. Luciferase signal was measured and IFN-β or ISG54/ISRE induction relative to EV was
calculated. Inhibition of interferon-stimulated gene (ISG) expression was analyzed in HEK-293T (C) cells using RT-
qPCR analysis. Cells were transfected with 250 ng of pCAGGS EV or pCAGGS vector encoding RVP or LBV-GH P.
After 24 h, cells were stimulated using 100 U/ml recombinant IFN, RNA was extracted 18 h after treatment, and the
expression of antiviral ISGs *CCL5*, *IFIT1*, and *MX1* was quantified. Relative quantification was performed using the $2^{-\Delta\Delta Ct}$ method and *TBP* as a reference gene. All experiments were performed in triplicate and mean values and SD were
calculated. **** = $p < 0.0001$; *** = $p < 0.001$; ** = $p < 0.01$; * = $p < 0.05$; ns = not significant; determined by student's t-
test (see S2 Table for details).

human and primate cells [14, 16–19, 57–59] encouraged us to further investigate specific lyssa-
virus P protein functions in *E. helvum* bat cells. For this, we established a bat-specific IFN-β
promoter-reporter assay. Five so far unknown sequences of the positive regulatory domains
(PRDs) of different bat IFN-β promoters, which represent the central hub for the regulation of
antiviral IFN-β transcription [60], were amplified, sequenced, and aligned with all available
bat sequences and representative sequences from selected mammalian taxa and *Homo sapiens*
(Fig 3A). GenBank entries were uploaded for the respective sequences (see Material and Meth-
ods). All applied Yinpterochiroptera PRDs were highly conserved except for the *Pteropus
alecto* (*P. alecto*) PRD that exhibited two nucleotide polymorphisms. Sequences from other
mammalian representatives showed a reduced nucleotide identity of <89% to the *E. helvum*
PRD. This finding is in line with our own previous experience that human IFN-β promoter
plasmids show limited activation in bat cell lines. For the generation of a bat-specific IFN-β
promoter activation reporter plasmid, applicable to multiple bat species, a representative IFN
promoter region (-278 to -89 upstream of ATG) from previously established *Rousettus aegyp-
tiacus* kidney cells (RoNi/7) [42] was cloned into the commercially available reporter vector
pGL4.10[luc2], encoding a genetically engineered firefly (FF) luciferase from *Photinus pyralis*.
Subsequently, the bat-specific pGL4-IFN-β reporter construct, designated IFN-β-FF, was
tested and titrated in *E. helvum* bat cells. To identify the optimal reference plasmid for the
detection of the baseline transcription factor activity in bat cells we tested different *Renilla
reniformis* (RL) luciferase-expressing plasmids controlled by an SV40 (pRL-SV40) or a thymi-
dine kinase promoter (pGL4.74 [hRLuc/TK]) (Fig 3B). Taking into account the luminometer
settings, the linear range of detection is achieved above 1,000 RLU with background counts
<100 RLU. Optimal RL luciferase read-out was therefore only achieved upon transfection of
25 ng pRL-SV40 plasmid with RLU values >1,000 (Fig 3B). IFN-β promoter activation was
inducible by VSV-RNA transfection and dose-dependent ranging from 0.1 to 0.4 relative
induction (FF/RL). Comparison of non-transfected with VSV-RNA transfected cells showed
that the fold induction was highest (94-fold) using 250 ng IFNb-FF, which was subsequently

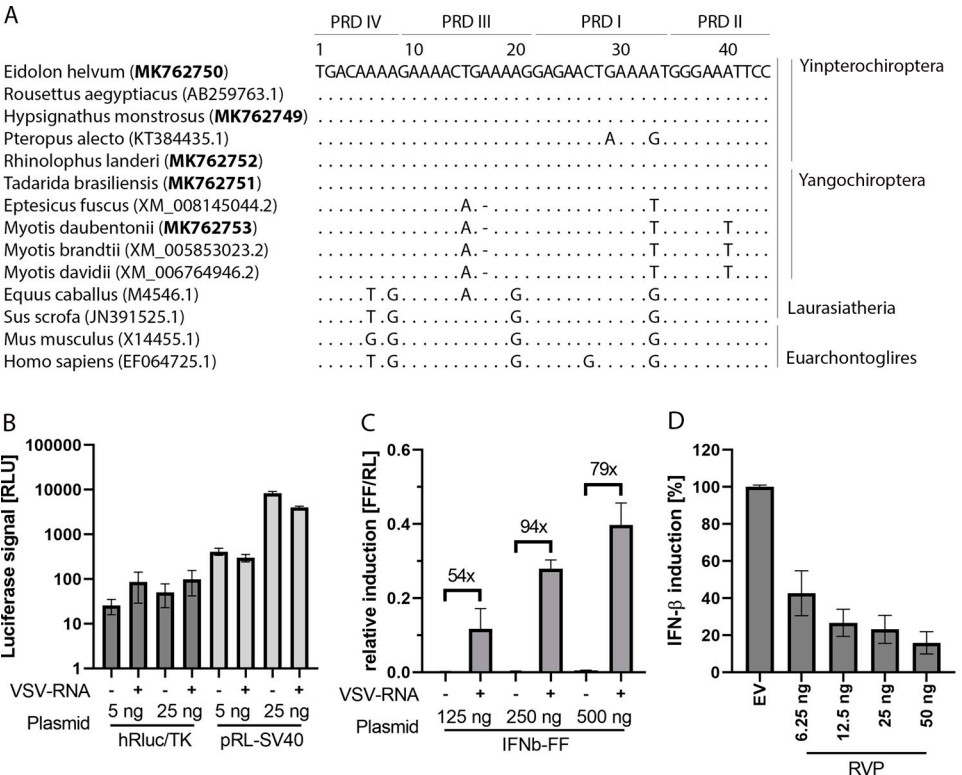

**Fig 3. Positive regulatory domains of IFN-β promoters in selected mammals.** Optimization of IFN-β promoter activation reporter assay in bat cells. (A) Positive regulatory domains (PRDs) of selected mammalian IFN-β promoters were aligned using Geneious 9.1.8 and sorted by taxonomy. Bold GenBank accession numbers indicate new sequences deposited as part of this paper. The PRDs of the *E. helvum* IFN-β promoter were set as a reference sequence. Dots indicate identical bases; dashes show missing nucleotides relative to the reference sequence. PRDs I to IV are indicated, representing binding sites for the transcription factors AP-1 (PRD IV), IRF3/IRF7 (PRD III and PRD I), and NF-κB (PRD II). (B) Titration of *Renilla* (RL) luciferase plasmids for IFN-β reporter assay. *E. helvum* lung cells (EidLu/20.2) were transfected with RL luciferase reporter plasmids as indicated (details in the methods section). Stimulation was achieved by transfection of 1 µg VSV-RNA (total RNA from vesicular stomatitis virus-infected cells). Cells were lysed after 18 h and RL luciferase signal was determined. (C) Titration of firefly (FF) luciferase vector by transfection of 25 ng pRL-SV40 plasmid and bat-specific pGL4-IFN-β reporter construct as indicated and stimulation with VSV-RNA. Results are presented as ratios (FF values/RL values). (D) EidLu/20.2 cells were transfected with 25 ng of pRL-SV40 and 250 ng of pGL4-IFN-β reporter plasmids and 50 ng of pCAGGS empty vector (EV) or increasing amounts of Rabies virus phosphoprotein (RVP) expression vector. Cells were stimulated by VSV-RNA transfection and lysed after 18 h. Luciferase signal was measured and IFN-β induction relative to EV was calculated. All experiments were performed in triplicate (C, D), or duplicate (B), and mean values and SD were calculated.

applied in follow-up experiments. To assess overall assay performance, we tested RVP-dependent inhibition of IFN-β promoter activation, which is a broadly active IFN antagonist. As shown in Fig 3D, IFN-β promoter activation was inhibited in a dose-dependent manner ranging from ~50% (6.25 ng pCAGGS-RVP) to ~90% inhibition (50 ng pCAGGS-RVP).

## IFN-ß inhibition by LBV P is conserved and highly efficient in bat cells

Next, we reassessed our findings on LBV P IFN-ß inhibitory activity in bat cells and performed IFN-β promoter activation reporter assays in *E. helvum* cells. To better reflect lyssavirus diversity, we included the P proteins of two additional LBV isolates (LBV-Nig56, lineage B, Nigeria, accession no. EU293110; LBV-Sen, lineage A, Senegal, EU293108) [61], representing lyssavirus phylogroup II, as well as Duvenhage virus (DUUV; accession no. EU293120) and street rabies

virus 1088 (RABV-1088; accession no. AB645847.1) P proteins as phylogroup I representatives.

To exclude cell line-specific effects, we compared the IFN-ß inhibitory activity of all mentioned lyssavirus P in two human cell lines (HEK-293T, A549) and two *E. helvum* cell lines (EidLu/20.2, EidNi/41.3). Representative immunofluorescence assays and control Western blots were performed to illustrate transfection efficiency and P protein expression in the analyzed cell lines (S2 Fig). Protein levels were normalized using actin as a reference protein, showing generally comparable P protein expression. All LBV P showed significantly lower inhibition of IFN-β promoter activation than phylogroup I representatives RVP, RVP 1088, and DUVV P in human cell lines (Fig 4A and 4B). Whereas RVP and DUVV P were able to reduce IFN-β promoter activation by ~95% (HEK-293T), and ~70% (A549), all applied LBV P variants exhibited significantly reduced efficiency in HEK-293T (73% to 86% inhibition), and A549 cells (42% to 46% inhibition). Noteworthy, RVP of the RABV-1088 street strain showed enhanced IFN-β inhibition in A549 cells, corroborating previous findings of Masatani et al. using a comparable luciferase-based approach [18]. In both bat cell lines, no significant decrease of IFN-β inhibition was detected between RVP and all tested LBV P constructs exhibiting a highly comparable inhibition of >87% (Fig 4C and 4D). In contrast to the other tested cell lines, all LBV P variants showed slightly enhanced IFN-ß inhibitory activity in comparison to RVP in EidNi/41.3 cells (Fig 4D). In sum, we show that, as compared to prototypic RVP, LBV P anti-IFN-β activity is substantially reduced in human cells but robust and comparable to other lyssavirus P in cells from the putative natural bat reservoir.

## Discussion

Bats host a plethora of viruses and seem to tolerate virus infections without showing severe clinical symptoms [29, 31]. This might be connected to an enhanced IFN response with reduced general inflammation upon virus infection, possibly facilitating virus persistence instead of acute infections [35, 62]. By using a newly established bat-specific IFN promoter activation assay we showed that LBV P has robust IFN-ß inhibitory activity in reservoir host-derived bat cells. In contrast, its capability to block IFN-ß induction was clearly reduced in human cell lines compared to the impact of prototypic RVP. The conserved and strong IFN-ß inhibitory activity of all tested LBV P proteins might be connected to long-term host adaptation processes to the bat cell environment [35]. In addition, a strong and continuously upregulated IFN response [34] would force bat-associated viruses to evolve highly active P proteins to prevent virus elimination. Although we did not observe the previously described [34] elevated basal transcriptional upregulation of *IFNB1* mRNAs in *E. helvum* cells, we detected an enhanced bat cell-specific IFN response upon transfection of dsRNA molecules and infection with an RVFV reporter virus lacking its IFN antagonist NSs.

A previous study showed differential IFN-ß inhibitory activity of lyssavirus P proteins in human cells [18]. Using human HEK-293T cell-based assays, Masatani and colleagues found that all lyssavirus P proteins block IFN induction via TBK-1. In addition, street strain RVP variants showed enhanced IKKepsilon (IKKε)-dependent IFN-ß inhibitory activity, as compared to cell culture-adapted RVP or other lyssavirus P proteins. The authors suggested that the additional function might inhibit the NF-κB-dependent proinflammatory response. We observed that LBV P proteins (phylogroup II) derived from LBV isolates from wild bats show reduced IFN-ß inhibition in HEK-293T and A549 cells as compared to prototypic RVP. The strongest IFN-ß inhibitory activity was indeed seen for the street strain variant 1088 P protein. SAG2 strain-based RVP and DUVV P protein also showed strong IFN-ß inhibitory activity in HEK-293T cells that was, however, not seen in A549 cells. An explanation might be that we induced

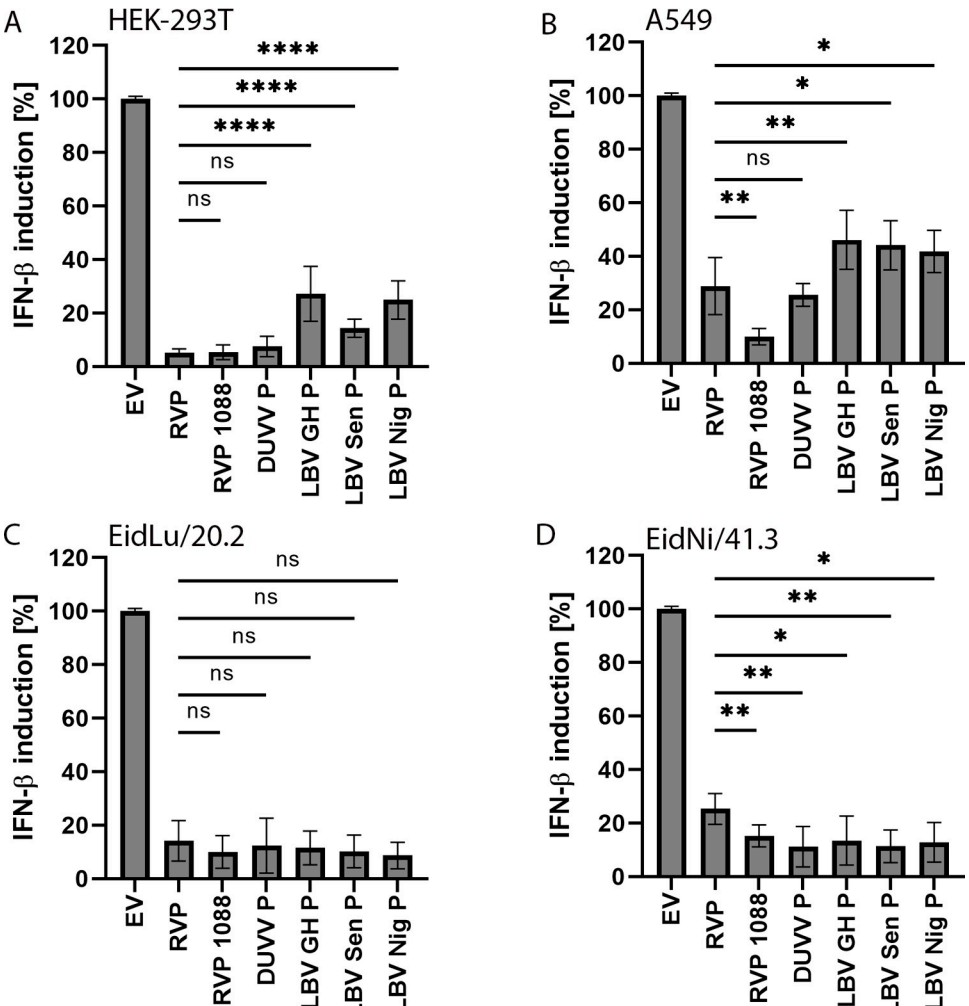

**Fig 4. Reduced LBV P-based inhibition of IFN-β promoter activation in human compared to bat cell lines.**
Human kidney (HEK-293T; A) and lung (A549; B), and *E. helvum* kidney (EidNi/41.3; D) and lung (EidLu/20.2; C)
cells were transfected with *Renilla* (RL) and firefly (FF) luciferase reporter plasmids, and 50 ng of pCAGGS empty
vector (EV) or following FLAG-tagged lyssavirus P proteins: Rabies virus P protein (RVP), RVP street rabies strain
1088 (RVP 1088), Duvenhage virus P protein (DUVV P), LBV Ghana isolate (LBV-GH), LBV Senegal isolate
(LBV-Sen) and LBV Nigeria isolate (LBV-Nig), respectively. IFN induction was achieved by transfection of 1 μg
VSV-RNA (total RNA from vesicular stomatitis virus-infected cells). Cells were lysed after 18 h and luciferase signal
was measured. Data were normalized using RL luciferase reference plasmid. For comparability between cell lines, IFN-
β induction relative to the EV negative control was determined. Experiments were performed in triplicate and mean
values from three independent experiments and SD are shown. **** = p<0.0001; ** = p<0.01; * = p<0.05; ns = not
significant; determined by student's t-test (see S2 Table for details).

the IFN response with total RNA from VSV-infected cells, known to trigger both RIG-I and
MDA5 helicases. Masatani et al. overexpressed specific signaling molecules, e.g. IRF-3, and
infected cells with New Castle Disease virus known to be sensed by RIG-I. In addition, the
authors applied different promoter activation plasmids focusing on IRF-3-dependent pro-
moter binding, whereas we used the complete IFN-beta promoter region that is targeted by
multiple transcription factors. An MDA5-triggered IFN induction might surpass the IKKε-
mediated IFN response and, for example, stimulate NF-κB-dependent IFN promoter activa-
tion. Of note, overexpression of signaling molecules as well as transfection of exogenous RNA

are both well-established methods to analyze immune pathways and highly sensitive and fine-tuned IFN-ß promoter reporter assays. While the use of such stimuli enables specific targeting of certain components of immune pathways, virus infections are less artificial and elicit more complex cellular responses [63]. During replication, VSV produces different RNA species that induce both RIG-I and MDA5, although RIG-I seems to be predominantly involved in generating an efficient antiviral immune response [64, 65]. Comparably, RIG-I was identified as the primary sensor molecule for rabies virus as prototypic representative of the virus family Rhabdoviridae, which also encompasses VSV [66]. In addition. rabies virus was also found to induce MDA5-dependent IFN signaling [11], whereas *in vivo* experiments suggested a superior role for RIG-I activation in survival and IFN production upon both rabies virus and VSV infection. While using exogenous RNA stimuli simplifies complex virus-host interactions and focuses on the relevant signaling pathways, detrimental effects of virus infection like transcriptional or translational shutoff are avoided [67, 68].

In bat cells, all lyssavirus P proteins of both phylogroups showed strong activity, possibly indicating that all P proteins interfere with bat TBK-1 and bat IKKε, blocking IFN-ß induction and NF-κB activation at the same time. This would support previous suggestions that all street strain lyssavirus P proteins originally have a double function that might be lost through cell culture passaging in fixed, culture-adapted lyssavirus strains. Detailed protein interaction and promoter activation studies with bat TBK-1 and bat IKKε should further investigate this hypothesis. In human cells, the induction of other IFN subtypes, such as IFN-α largely relies on the same signaling molecules as IFN-ß, including TBK-1 and IKKε, but does not require NF-κB signals for gene expression [69]. IFN-α is not as well characterized as IFN-ß in the context of rabies virus infection and seems to play a subordinate role in antiviral signaling [70] and was therefore not assessed in this study. Due to signaling through TBK-1 and IKKε, the major targets of RVP interference, a comparable RVP-based inhibition of IFN-α promoter activation is highly likely. Interestingly, IFN-λ has very recently been shown to attenuate rabies virus infection and induce antiviral genes, although P protein-based interference with the induction of distinct IFN classes is less well described [71].

By using an ISG54/ISRE promoter activation assay, we observed a slightly increased capacity of LBV P as compared to RVP to block IFN signaling in both human and bat cells using a human ISRE reporter construct. While this increase in efficiency might translate to an increased replicative capacity of LBV over rabies virus, we were not able to detect differences between RVP and LBV P in inhibiting ISG expression in human cells. In contrast, the decreased IFN-ß inhibitory action of LBV P was only detectable in human cells, highlighting the need for species-specific tools to evaluate antiviral immune responses. Nevertheless, future studies should not only further evaluate the consequences of P protein activity in bat cells in the context of lyssavirus infection but also evaluate the capacity of P proteins to block JAK-STAT signaling in bat cell systems with appropriate bat-specific assays.

To strengthen our findings on IFN-ß inhibition, we not only included two established human cell lines but also distinct *E. helvum* bat cells lines in the described IFN-ß promoter activation studies. Although these cell cultures are each derived from different organs (lung, kidney), observing cell line specific effects cannot be completely ruled out. In addition, the order Chiroptera is highly diverse and general conclusions on bat immune responses are difficult to draw. As we focused on LBV in this study, we chose cells from the putative reservoir host *E. helvum* for our analyses. However, it has to be noted that the described findings might not hold true for cell lines from other species of this large mammalian taxon. A study by Hölzer et al. has recently analyzed the IFN-induced transcriptome of Yangochiroptera bat cells in great detail, revealing substantial differences to published findings on Yinpterochiroptera bats [55]. The authors not only describe the lack of multiple ISGs that were described in

Yinpterochiroptera bats but also failed to confirm earlier findings of IFN-α gene expansion or high baseline IFN expression in bat cell lines. These discrepancies further highlight the need for bat-specific tools and an overall caution regarding broad and general statements on bat immune responses. Future studies should not only consider multiple bat species but also implement an increased repertoire of cell culture models to further elucidate the interaction of lyssaviruses and the immune system of distinct bat representatives.

Noteworthy, the impact of IFN antagonists on virus replication and IFN signaling in different bat cells is still not fully understood and few studies have addressed this topic [72–74]. In the case of bats, it is largely unknown to what extent the immune system influences the manifestation of viral diseases or their progression. However, recent research suggests that bats have evolved unique features to fight RNA viruses [75]. One of the most important findings is the discovery of an attenuated inflammatory response in bats [33, 76] that is tightly connected to the IFN response. While Ahn et al. reported a dampened inflammasome activation in bat cells, others suggested enhanced virus restriction based on elevated metabolism or a particularly effective IFN response [33, 77, 78]. To analyze the extent of differential LBV P activity on viral replication and proinflammatory responses would rely on reverse genetics approaches and more bat-specific tools to investigate proinflammation in the context of virus infection.

Other studies also reported high ISG expression in unstimulated bat cells, possibly enhancing intrinsic virus resistance [79]. A recent study by Irving et al. elegantly describes the influence of interferon regulatory factors 1/3/7 on ISG expression in bat cells [80]. Besides showing high expression levels of these central antiviral regulators in bat cells and tissue, which translates to elevated ISG expression, the authors also describe a prolonged IFN-like antiviral signature and novel bat-specific antiviral mechanisms. In light of these findings, developing bat-specific tools to integrate IFN induction pathways, on the one hand, and expression of ISGs as antiviral effectors, on the other hand, might be an important step towards decoding the unique antiviral immune response of bats.

In sum, our established bat-specific IFN-β promoter-reporter assay might serve as an *in vitro* risk assessment tool to analyze bat virus-derived IFN antagonists in bat cells. This approach enables comparisons with other mammalian cell systems to explore distinct aspects of virus virulence and understand specific features of the bat IFN system [18, 81–83].

## Supporting information

**S1 Fig. Interferon competence of selected cell lines and ISG54/ISRE activation in bat cells.** The functionality of IFN signaling pathways was analyzed in A549 (A) and EidLu/20.2 (B) cells. After seeding cells, increasing amounts of total RNA from Vesicular stomatitis virus (VSV)-infected cells were transfected as indicated. After 20 h the supernatants were collected, diluted as indicated, and added to naïve cells of the respective cell line. After 7 h cells were infected with RVFV-Ren (MOI 0.002), incubated for 16 h, lysed and luciferase signal was measured. Virus growth was calculated relative to the mock-treated cells (0 μg) for the respective dilution. High values indicate increased virus replication and low amounts of bioactive IFN in the supernatants. Experiments were performed in triplicate and mean values and SD are shown. The experimental procedure is depicted in (C). Upon detection of VSV-RNA type I IFNs are produced and secreted. Incubation of cells with IFN-containing supernatants induces an antiviral state characterized by the transcription of interferon-stimulated genes (ISGs). RVFV-Ren reporter virus growth is inhibited by various ISG proteins in IFN stimulated cells. EidLu/20.2 cells were transfected with RL and FF luciferase reporter plasmids (pISG54-luc) and 50 ng of pCAGGS empty vector (EV) or pCAGGS vector encoding Rabies virus P protein (RVP) or LBV-GH P protein (LBV GH P). Cells were stimulated by recombinant IFN

treatment (100 U/ml) and lysed after 18 h. Luciferase signal was measured and ISG54/ISRE induction relative to EV was calculated (D). All experiments were performed in triplicate and mean values and SD were calculated. $^*$ = $p < 0.05$; determined by student's t-test (see S2 Table for details).
(TIF)

**S2 Fig. Expression of FLAG-tagged lyssavirus P proteins in human and bat cells.** Human kidney (HEK-293T) and *E. helvum* lung (EidLu/20.2) cells were transfected with pCAGGS vector encoding P proteins from human- and bat-associated lyssaviruses (Rabies virus P protein (RVP); RVP street rabies strain 1088 (RVP 1088); Duvenhage virus P protein (DUVV P); LBV Ghana isolate P protein (LBV-GH P); LBV Senegal isolate P protein (LBV-Sen P); LBV Nigeria isolate P protein (LBV-Nig P)). (A) For immunofluorescence assay cells were seeded on glass coverslips in 24-well plates and transfected with 0.5 µg plasmid DNA per well. Cell fixation and lysis were performed 24 h after transfection. For immunofluorescence microscopy, cells were stained with mouse-anti-FLAG antibody (1:100) and goat anti-mouse-Cy3 antibody (1:200). Cellular DNA was stained with DAPI. Bars represent 50 µm. (B) For Western blot analysis, cell lysates from transfected (2 µg/6-well, respectively) HEK-293T, and A549, EidLu/20.2, and EidNi/41.3 were prepared 48 h post-transfection with Pierce RIPA lysis buffer (Thermo Fisher). Protein lysates and Spectra™ Multicolor Broad Range Protein Ladder (Thermo Fisher) were separated by SDS-PAGE using 10% polyacrylamide gels (120 V, 45 min). Western blot analysis was performed with mouse-anti-FLAG (1:3,000; Sigma-Aldrich) and mouse-anti-β-actin (1:5,000; Sigma-Aldrich) primary antibodies and fluorescently-labeled goat-anti-mouse (1:15,000; 800 nm; Licor), or goat-anti-rabbit (1:15,000; 680 nm; Licor) secondary antibody. Fluorescent signal was visualized with the Licor Odyssey CLx imaging system (Licor), fluorescent signal intensities were measured and FLAG-based signal intensity was normalized to actin reference protein using the Image Studio™ software package.
(TIF)

**S1 Table. PCR primers and probes.**
(DOCX)

**S2 Table. Results of unpaired student's t-tests.**
(DOCX)

**S1 File. Raw Western blot images.**
(PDF)

## Acknowledgments

We thank Nicolas Heinemann, Max Tschischka, Christina Metzger for excellent technical assistance and Dr. Sebastian Hauka for technical and scientific advice. We also thank the Friedrich-Löffler-Institute for providing the LBV-GH isolate, Prof. Klaus Conzelmann for the RVP plasmid, Prof. Friedemann Weber for sharing the RVFV-Ren reporter virus, and Prof. Takashi Fujita and Prof. David E. Levy for kindly sharing the p125-luc and ISG54-luc plasmids.

## Author Contributions

**Conceptualization:** Jan Papies, Andrea Sieberg, Marcel A. Müller.

**Data curation:** Jan Papies.

**Formal analysis:** Jan Papies, Andrea Sieberg.

**Funding acquisition:** Christian Drosten, Marcel A. Müller.

**Investigation:** Jan Papies, Andrea Sieberg, Daniel Ritz.

**Methodology:** Jan Papies.

**Project administration:** Christian Drosten, Marcel A. Müller.

**Supervision:** Daniela Niemeyer, Christian Drosten, Marcel A. Müller.

**Visualization:** Jan Papies.

**Writing – original draft:** Jan Papies, Andrea Sieberg, Marcel A. Müller.

**Writing – review & editing:** Jan Papies, Andrea Sieberg, Daniela Niemeyer, Christian Drosten, Marcel A. Müller.

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
