## [Decision Letter · Decision Letter 0]

31 Aug 2021

PONE-D-21-25746

Elevated anti-interferon activity of lyssavirus phosphoproteins in bat compared to human cells

PLOS ONE

Dear Dr. Müller,

Thank you for submitting your manuscript to PLOS ONE. After careful consideration, we feel that it has merit but does not fully meet PLOS ONE’s publication criteria as it currently stands. Therefore, we invite you to submit a revised version of the manuscript that addresses the points raised by two experts during the review process. These points include, but are not limited to, clarification and quantification of P protein Western blotting results and clearer justification of the authors' main conclusion (higher anti-interferon activity of P proteins in bat relative to human cells).

We look forward to receiving your revised manuscript.

Kind regards,

Michael Nevels

Academic Editor

PLOS ONE

Journal Requirements:

When submitting your revision, we need you to address these additional requirements. 1. Please ensure that your manuscript meets PLOS ONE's style requirements, including those for file naming. The PLOS ONE style templates can be found at https://journals.plos.org/plosone/s/file?id=wjVg/PLOSOne_formatting_sample_main_body.pdf and https://journals.plos.org/plosone/s/file?id=ba62/PLOSOne_formatting_sample_title_authors_affiliations.pdf
 2. During our internal checks, the in-house editorial staff noted that you conducted research or obtained samples in another country. Please check the relevant national regulations and laws applying to foreign researchers and state whether you obtained the required permits and approvals. Please address this in your ethics statement in both the manuscript and submission information. In addition, please ensure that you have suitably acknowledged the contributions of any local collaborators involved in this work in your authorship list and/or Acknowledgements. Authorship criteria is based on the International Committee of Medical Journal Editors (ICMJE) Uniform Requirements for Manuscripts Submitted to Biomedical Journals - for further information please see here: https://journals.plos.org/plosone/s/authorship.  In addition, to ensure that your manuscript complies with our policies on animal research, please specify if the permits permit no. CHRPE49/09; A04957 cover the method of capture and euthanasia used. 3. We note that you have stated that you will provide repository information for your data at acceptance. Should your manuscript be accepted for publication, we will hold it until you provide the relevant accession numbers or DOIs necessary to access your data. If you wish to make changes to your Data Availability statement, please describe these changes in your cover letter and we will update your Data Availability statement to reflect the information you provide. 4. We note that you have included the phrase “data not shown” in your manuscript. Unfortunately, this does not meet our data sharing requirements. PLOS does not permit references to inaccessible data. We require that authors provide all relevant data within the paper, Supporting Information files, or in an acceptable, public repository. Please add a citation to support this phrase or upload the data that corresponds with these findings to a stable repository (such as Figshare or Dryad) and provide and URLs, DOIs, or accession numbers that may be used to access these data. Or, if the data are not a core part of the research being presented in your study, we ask that you remove the phrase that refers to these data. 5. PLOS ONE now requires that authors provide the original uncropped and unadjusted images underlying all blot or gel results reported in a submission’s figures or Supporting Information files. This policy and the journal’s other requirements for blot/gel reporting and figure preparation are described in detail at https://journals.plos.org/plosone/s/figures#loc-blot-and-gel-reporting-requirements and https://journals.plos.org/plosone/s/figures#loc-preparing-figures-from-image-files. When you submit your revised manuscript, please ensure that your figures adhere fully to these guidelines and provide the original underlying images for all blot or gel data reported in your submission. See the following link for instructions on providing the original image data: https://journals.plos.org/plosone/s/figures#loc-original-images-for-blots-and-gels.   In your cover letter, please note whether your blot/gel image data are in Supporting Information or posted at a public data repository, provide the repository URL if relevant, and provide specific details as to which raw blot/gel images, if any, are not available. Email us at plosone@plos.org if you have any questions.

Reviewers' comments:

Reviewer's Responses to Questions

**Comments to the Author**

1. Is the manuscript technically sound, and do the data support the conclusions?

Reviewer #1: Partly

Reviewer #2: Partly

2. Has the statistical analysis been performed appropriately and rigorously? 

Reviewer #1: Yes

Reviewer #2: Yes

3. Have the authors made all data underlying the findings in their manuscript fully available?

Reviewer #1: Yes

Reviewer #2: Yes

4. Is the manuscript presented in an intelligible fashion and written in standard English?

Reviewer #1: Yes

Reviewer #2: Yes

5. Review Comments to the Author

Reviewer #1: PONE-D-21-25746, Papies et al.: Elevated anti-interferon activity of lyssavirus phosphoproteins in bat compared to human cells

Bats were reported to carry highly pathogenic zoonotic viruses often without showing symptoms or suffering from disease. This is largely attributed to an enhanced (IFN) antiviral and decreased inflammatory response in bats. In the present work, the authors address the question whether the phosphoprotein (P) of Lagos bat lyssavirus (LBV), a rabies-related virus, has an enhanced anti-IFN activity in cells of their natural bat host species as compared to human cells. This might indicate adaptation to the enhanced antiviral response in bats.

To this end, they generated Eidolon bat cell lines and novel bat IFN-ß reporter plasmids and provide their characterization, including IFN-ß response and permissivity of cells for a LBV isolate (Fig. 1, S1), and response of the novel plasmids in bat cells to RNA- and viral PAMPs (Fig. 3). In reporter gene experiments – and somewhat contrary to the title - they observe roughly comparable inhibition of human and bat IFN-ß promoter activation by control rabies virus P proteins, while the inhibitory activity of LBV P protein in human cells is markedly reduced and, if at all, only marginally better in bat cells (significant in 1 of 2 cell lines)(Fig. 4).

While most of the provided experiments were performed and presented adequately, I do not agree with the conclusions emphasized in the title and discussion of “enhanced anti-interferon activity of lyssavirus phosphoproteins in bat compared to human cells”.

In my opinion the data indicate that the activity of LBV P (and only with respect to IFN-ß promoter activation) is not enhanced in bat cells, but rather reduced in human cells. Note that RABV P and DUVV, which is another bat lyssavirus, perform well in both human and bat cells, the exception is LBV P showing defects in human cells.

Major:

1. Apart from the “enhanced in … bat” issue, the title is too general and broad, it should be limited to: Elevated IFN-ß inhibitory (not: anti-interferon) activity of activity of Lagos Bat lyssavirus phosphoproteins in bat compared to human cells. Note that DUVV is also a bat lyssavirus, and RV a lyssavirus, and the present title would misleadingly implicate a common general feature of lyssaviruses. In addition, only IFN-ß promoter activation was studies in bat cells, not anti-IFN activity.

2. With respect to a general “anti-interferon” activity as implied in title, Fig. 2 is the most meaningful. There it is shown in human cells that LBV P seems to have deficits in IFN-ß promoter inhibition (Fig. 2A), while it catches up or even overtakes RVP when it comes to ISRE inhibition (Fig. 2B), i.e., regarding IFN induction and IFN-STAT signaling is regarded in total. Enhanced STAT inhibition would be very exciting and could be determined in a simple and specific ISRE experiment, just by adding exogenous IFN rather than transfect RNA. In bat cells, if bat ISRE reporters are not available, RT-PCR for ISGs could be done. Such simple experiment would clarify whether LBV P is better adapted to antagonize bat anti-interferon responses, or not.

3. S2B Fig.: There seems to be something wrong with the P proteins in WB. First, RVP is of different size in HEK-293T (<<40 kDa) and other cell lines (>=40 kDa): check marker. More worrying is the identity of the proteins, as in HEK-293T and EidNi/41.3 LBV-GH-P is smaller than the other lyssavirus P proteins, while in A549 and EidLu/20.2 the small protein is LBV-Nig P. On first sight, it appears that LBV-GH and LBV-Nig proteins were interchanged. Please check, and make sure correct designation in the other experiments and tables. Although this does not severely affect the overall conclusion, it is not suitable to inspire confidence in the experimental diligence.

4. S1 Fig. IFN competence and antiviral activity of IFN in human A549 (A) and EidLu/20.2 bat cells (B). Unexpectedly, and in contrast to the popular view, IFN bioactivity is way lower in bat cells. Irrespective of this, however, the somewhat (2-fold) lower replication of LBV in A594 (Fig. 1) is not attributed to the way higher bioactivity of A549, but rather to an allegedly higher LBV anti-IFN activity in EidLu/20.2. Please reconsider such reasoning.

In addition, in Fig. 1 please include LBV infectious titers, which are more relevant for appreciation of the full extent of antiviral activity than genome copies.

Minor:

l. 383ff: move promoter sequence IDs to Materials and Methods, in addition, provide sequence IDs of P proteins used.

l.47: replace frequently with sporadically (human transmissions)

l. 388: Yinpterochiroptera is mentioned in the text, please indicate suborder in Fig. 3

l. 412: replace Fig 3C with Fig 3D

Reviewer #2: The authors outline their new luciferase induction model for analysis of IFN-beta kinetics in bat cells. The details are clearly worked out and explained with some key points that are relevant to the field. The claims based on differences in IFN induction between different RVP/LBV constructs are based on western blot images the authors claim are equally expressed. P is clearly differentially expressed between constructs and cell lines and these needs to be quantified and normalized to housekeeping before any claim on differences can be made. It looks like there are large differences in LVP vs RVP expression that would affect these results.

For figure 4 internally there are statistical differences between constructs within the cell line but in the results the authors also compare results across cell lines. To prove this is relevant/significant the data needs to be directly compared with statistics between bat vs human for example. It is good they have two bat and two human cell lines but the differences are still quite small and ideally a third bat and human cell line should be included. Failing that, the authors need to clearly state that with such a limited number of cell lines all affects could simply be cell-line specific and not related to species differences, highlighting the limitations in the study.

While the authors carefully worked out promoter/concentration differences in the EiD cell lines the authors need to mention in the discussion how this is all done with artificial stimulation (transfected rna / rvfv reporter etc) and how it may differ with real infection. They should also indicate references etc showing how LBV/RBV infection would be expected to activate the same RIG-I/MDA-5 signaling pathways as VSV RNA.

There is also limited discussion on the differences between IFNs and if other (for example alpha, omega, kappa) promoters are expected to be affected in a similar fashion. There was limited discussion on ISG54 promoter whereby no species differences were observed and how what may be more relevant during infection - P's effect on IFN-b promoter compared to direct on ISG induction visa ISRE elements. (a comparison of ISG54 induction in the bat cells would be ideal).

With that in mind there are other studies showing high ISG induction and characterizing IFN induction (the authors own work, ref. 56 and also https://pubmed.ncbi.nlm.nih.gov/33147460/) that are highly relevant to this study and should be discussed. Additionally there is data showing unique bat-specific IRf3 phospho sites that may be relevant to activation (and viral-induced inhibition from TBK1) that could be discussed - https://www.cell.com/iscience/fulltext/S2589-0042(20)30142-5.

6. PLOS authors have the option to publish the peer review history of their article (what does this mean?). If published, this will include your full peer review and any attached files.

Reviewer #1: No

Reviewer #2: No

---

## [Author Response · Author response to Decision Letter 0]

13 Dec 2021

Reviewer #1: PONE-D-21-25746, Papies et al.: Elevated anti-interferon activity of lyssavirus phosphoproteins in bat compared to human cells

Bats were reported to carry highly pathogenic zoonotic viruses often without showing symptoms or suffering from disease. This is largely attributed to an enhanced (IFN) antiviral and decreased inflammatory response in bats. In the present work, the authors address the question whether the phosphoprotein (P) of Lagos bat lyssavirus (LBV), a rabies-related virus, has an enhanced anti-IFN activity in cells of their natural bat host species as compared to human cells. This might indicate adaptation to the enhanced antiviral response in bats.

To this end, they generated Eidolon bat cell lines and novel bat IFN-ß reporter plasmids and provide their characterization, including IFN-ß response and permissivity of cells for a LBV isolate (Fig. 1, S1), and response of the novel plasmids in bat cells to RNA- and viral PAMPs (Fig. 3). In reporter gene experiments – and somewhat contrary to the title - they observe roughly comparable inhibition of human and bat IFN-ß promoter activation by control rabies virus P proteins, while the inhibitory activity of LBV P protein in human cells is markedly reduced and, if at all, only marginally better in bat cells (significant in 1 of 2 cell lines)(Fig. 4).

While most of the provided experiments were performed and presented adequately, I do not agree with the conclusions emphasized in the title and discussion of “enhanced anti-interferon activity of lyssavirus phosphoproteins in bat compared to human cells”.

In my opinion the data indicate that the activity of LBV P (and only with respect to IFN-ß promoter activation) is not enhanced in bat cells, but rather reduced in human cells. Note that RABV P and DUVV, which is another bat lyssavirus, perform well in both human and bat cells, the exception is LBV P showing defects in human cells.

Major:

1. Apart from the “enhanced in … bat” issue, the title is too general and broad, it should be limited to: Elevated IFN-ß inhibitory (not: anti-interferon) activity of activity of Lagos Bat lyssavirus phosphoproteins in bat compared to human cells. Note that DUVV is also a bat lyssavirus, and RV a lyssavirus, and the present title would misleadingly implicate a common general feature of lyssaviruses. In addition, only IFN-ß promoter activation was studies in bat cells, not anti-IFN activity.

(new line numbers refer to the revised manuscript file with track changes)

Reply: We agree with the reviewer’s remark regarding the title of the manuscript and modified it to “Reduced IFN-ß inhibitory activity of Lagos bat virus phosphoproteins in human compared to Eidolon helvum bat cells” 

Additional changes were implemented to comply with the reviewer´s request:

Lines 37-41: “The same pattern was seen for two prototypic LBV P variants suggesting an overall enhanced anti-IFN activity E. helvum bat cells compared to human cells.” was changed to (lines 41-44) “The same pattern was seen for two prototypic LBV P variants suggesting an overall reduced LBV P IFN-ß inhibitory activity in human cells as compared to E. helvum bat cells.”

“…anti-IFN…” was changed to “…IFN-ß inhibitory…”(lines 30, 35, 42, 116, 508, 515, 542, 576-577, 578, 586, 590, 594, 617, 675)

 “…anti-IFN activity…” was changed to “…IFN-ß inhibition…” (lines 43, 506, 593, 676)

Lines 465-466: “Enhanced inhibition of IFN-β promoter activation by lyssavirus P proteins in bat cell lines” was changed to “Reduced LBV P-based inhibition of IFN-β promoter activation in human compared to bat cell lines” (lines 547-548)

Line 438: “Anti-IFN function of…” was changed to “IFN-ß inhibition by…” (line 506)

Line 457, 459: „…IFN inhibition“ was changed to „…IFN-ß inhibition…” (lines 593, 676)

Lines 484-485: ” By using a newly established bat-specific IFN promoter activation assay we showed that LBV P proteins have enhanced IFN-ß inhibitory activity in reservoir host-derived bat cells as compared to human cells.“ was changed to “By using a newly established bat-specific IFN promoter activation assay we showed that LBV P proteins have robust IFN-ß inhibitory activity in reservoir host-derived bat cells. In contrast, IFN-ß was clearly reduced in human cell lines.” (lines 574-576)

2. With respect to a general “anti-interferon” activity as implied in title, Fig. 2 is the most meaningful. There it is shown in human cells that LBV P seems to have deficits in IFN-ß promoter inhibition (Fig. 2A), while it catches up or even overtakes RVP when it comes to ISRE inhibition (Fig. 2B), i.e., regarding IFN induction and IFN-STAT signaling is regarded in total. Enhanced STAT inhibition would be very exciting and could be determined in a simple and specific ISRE experiment, just by adding exogenous IFN rather than transfect RNA. In bat cells, if bat ISRE reporters are not available, RT-PCR for ISGs could be done. Such simple experiment would clarify whether LBV P is better adapted to antagonize bat anti-interferon responses, or not.

Reply: In the underlying experiments of Fig. 2B, C we actually added exogenous type I IFN and did not transfect RNA molecules. To avoid such misunderstandings we now added the applied stimulus in the figure itself (see revised Figure 2). 

As suggested by the reviewer, we performed ISRE-Luc reporter assays with exogenous IFN stimulation in E. helvum cells using an available human ISG54/ISRE reporter construct (new S1 Fig C). Comparable to our results in the human cell line, LBV P showed a slightly increased capacity to block ISRE promoter activation as compared to RVP. A limitation might be that we applied a human ISRE reporter plasmid. However, ISRE motifs are conserved among mammals and we observed a robust ISG54/ISRE induction in the E. helvum cell line upon IFN treatment confirming the general integrity of the assay. 

Revised S1 Fig:

In addition, we performed RT-PCRs for ISGs after stimulating HEK293T cells expressing RVP or LBV P with exogenous type I IFN, as suggested by Reviewer 1. The results presented in revised Fig. 2C show a significant reduction of CCL5, IFIT1, and MX1 expression in both RVP and LBV P expressing cells as compared to empty vector control. In contrast to the ISRE reporter luciferase assay, LBV P did not show a significantly stronger inhibition of ISG transcription than RVP. More detailed analyses would be necessary to follow up on this observation, but were clearly not within the scope of the current study, which focuses on the differences in IFN-ß promoter activation.

Revised Fig. 2

3. S2B Fig.: There seems to be something wrong with the P proteins in WB. First, RVP is of different size in HEK-293T (<<40 kDa) and other cell lines (>=40 kDa): check marker. More worrying is the identity of the proteins, as in HEK-293T and EidNi/41.3 LBV-GH-P is smaller than the other lyssavirus P proteins, while in A549 and EidLu/20.2 the small protein is LBV-Nig P. On first sight, it appears that LBV-GH and LBV-Nig proteins were interchanged. Please check, and make sure correct designation in the other experiments and tables. Although this does not severely affect the overall conclusion, it is not suitable to inspire confidence in the experimental diligence.

Reply: We thank the reviewer for his/her observation. First, we sequenced all previously applied plasmids and confirmed the correctness of the inserted lyssavirus P coding sequences. Next, we repeated all relevant transfection-based experiments and Western blot analyses to guarantee correct protein identity, loading, and experimental diligence. Please find the revised Western blot data, including a quantitative analysis based on reference protein normalization using fluorescence Western blotting in the updated S2B Fig. We now confirm that all applied P proteins had comparable expression levels upon transfection of the same amounts of plasmids. The previous discrepancy might have occurred due to sample loading issues or inefficient blotting procedures. Interestingly, we again found that RVP 1088 had a minor deviation regarding the expected molecular weight exclusively in the two kidney cell lines (HEK-293T and EidNi/41.3). As the expected molecular weight was observed in both lung cells lines (A549, EidLu) we are convinced that the plasmid-driven protein expression is flawless but that some cell line specific modification of the P protein might lead to an apparent lower molecular weight. We cannot fully explain this effect, but differential phosphorylation patterns might explain this as the P proteins are known to be phosphorylated by unique protein kinases, resulting in altered P protein gel migration [1, 2]. Conceivably, different cell lines may encode distinct cellular protein kinases, possibly affecting P protein gel mobility depending on lyssavirus strain and cell line origin [1]. As RVP 1088 P protein shows substantial amino acid sequence variation compared to cell culture-adapted reference RVP, differential posttranslational modification appears to be a plausible explanation for the observed differences in gel mobility.

 

Revised S2 Fig

4. S1 Fig. IFN competence and antiviral activity of IFN in human A549 (A) and EidLu/20.2 bat cells (B). Unexpectedly, and in contrast to the popular view, IFN bioactivity is way lower in bat cells. Irrespective of this, however, the somewhat (2-fold) lower replication of LBV in A594 (Fig. 1) is not attributed to the way higher bioactivity of A549, but rather to an allegedly higher LBV anti-IFN activity in EidLu/20.2. Please reconsider such reasoning.

In addition, in Fig. 1 please include LBV infectious titers, which are more relevant for appreciation of the full extent of antiviral activity than genome copies.

Reply: We updated the relevant passages in the manuscript, considering the impact of reduced IFN bioactivity in E. helvum cells, as the reviewer correctly pointed out. 

Lines 331-333: “The higher final titers of LBV-GH in EidLu/20.2 might be linked to efficient virus-dependent control of the cellular antiviral response.” was changed to “The higher final titers of LBV-GH in EidLu/20.2 might be linked to the observed reduction of IFN bioactivity in this cell line. Another factor possibly affecting LBV propagation and RNA levels in human and bat cells might be virus-dependent control of the cellular antiviral response.” (lines 360-362).

We think that a decrease in LBV replication in A549 cells likely results from a combination of factors, including differential IFN bioactivity, LBV P impact on the IFN response, and possibly additional species-specific differences that were not addressed in our study. 

Regarding LBV infectious titers, we feel that RT-qPCR using fluorescent hydrolysis probes is an adequate method to quantify lyssavirus replication with high sensitivity and repeatability, which is frequently applied in scientific studies (see Faye et al., DOI:10.1016/j.jviromet.2016.12.019; Correia Moreira et al., DOI: 10.1016/j.jviromet.2019.04.025; Hughes et al., DOI:10.1128/JCM.42.1.299-306.2004; Pulmanausahakul et al., DOI:10.1128/JVI.02327-07.) In favor of a timely revision and considering the extraordinary capacity limitations in our BSL3 facility due to SARS-CoV-2/COVID-19-related work, we decided against repeating lyssavirus infection experiments to report infectious titers in the revised manuscript. Considering the nature of the discussed experiment and its role in the submitted manuscript, we think that taking this approach does not impact the outcome of our experiments or the conclusions we draw. 

Minor:

l. 383ff: move promoter sequence IDs to Materials and Methods, in addition, provide sequence IDs of P proteins used.

Reply: Promoter and P protein IDs were moved to Materials and Methods. P protein IDs were included in Material and Methods section (lines 237-239; 279-284).

l.47: replace frequently with sporadically (human transmissions)

Reply: “Sporadically” was inserted instead of “frequently”.

l. 388: Yinpterochiroptera is mentioned in the text, please indicate suborder in Fig. 3

Reply: Figure 3 was updated and complemented by bat suborders.

l. 412: replace Fig 3C with Fig 3D

Reply: Updated to Fig. 3D

Reviewer #2: The authors outline their new luciferase induction model for analysis of IFN-beta kinetics in bat cells. The details are clearly worked out and explained with some key points that are relevant to the field. The claims based on differences in IFN induction between different RVP/LBV constructs are based on western blot images the authors claim are equally expressed. P is clearly differentially expressed between constructs and cell lines and these needs to be quantified and normalized to housekeeping before any claim on differences can be made. It looks like there are large differences in LVP vs RVP expression that would affect these results.

Reply: We agree with the reviewer that the previous Western blot analysis was suboptimal and might have experienced some technical issues (e.g. sample loading). Therefore, after confirming the integrity of all applied plasmids we repeated the Western blots and complemented the analysis by quantification based on fluorescence Western blotting (please also see major comment 3 from Reviewer 1 (revised S2 Fig). In addition, we want to point out, that in Fig. 3D we showed that reducing RVP plasmid amounts by up to 75% did not significantly affect the reported IFN-β inhibition, indicating that slight variations in protein expression would not substantially alter the IFN-β inhibitory activity in our experimental setting. We show significant differences between RVP and LBV P IFN-β inhibitory activity using three different plasmids, derived from distinct LBV isolates. Also, we repeated these experiments three times and independently confirmed our findings in an additional cell lines for each species (human and E. helvum, three independent experiments, each with n=3). The significant difference in activity between RVP and LBV P in human cells is consistent in both analyzed cell lines for all three LBV P plasmids. In contrast, there was no significant loss of LBV P activity compared to RVP activity in both bat cell lines. We are therefore confident that our IFN-β promoter activation results were not affected by protein expression differences. 

For figure 4 internally there are statistical differences between constructs within the cell line but in the results the authors also compare results across cell lines. To prove this is relevant/significant the data needs to be directly compared with statistics between bat vs human for example. 

Reply: We avoid direct cross-species comparisons between individual P proteins in the revised manuscript. We fully agree with the reviewer that these can be misleading, considering cell line-specific effects and differential magnitudes of IFN-β induction. We believe that an independent comparison between LBV P and the well-described prototypic RVP (in each cell line individually) is more relevant for our analysis. This supports our claim that LBV P-based IFN-β inhibition is reduced in human cells (relative to RVP-based inhibition in the same cell line and experiment) compared to bat cells with respect to the well-established RVP effects in these cells. 

It is good they have two bat and two human cell lines but the differences are still quite small and ideally a third bat and human cell line should be included. Failing that, the authors need to clearly state that with such a limited number of cell lines all affects could simply be cell-line specific and not related to species differences, highlighting the limitations in the study. 

Reply: We agree that including more cell lines generally strengthens studies that compare cells from different species. We also understand that our experimental approach has limitations and stated these more clearly in the discussion. Although, we are confident that our approach using two distinct E. helvum cell lines from different organs and confirming our findings in three independent experiments per cell line, strongly reduces the risk of observing cell-line specific effects 

Lines 607-625: “To strengthen our findings on IFN-ß inhibition, we not only included two established human cell lines but also distinct E. helvum bat cells lines in the described IFN-ß promoter activation studies. Although these cell cultures are each derived from different organs (lung, kidney), observing cell line specific effects cannot be completely ruled out. In addition, the order Chiroptera is highly diverse and general conclusions on bat immune responses are difficult to draw. As we focused on LBV in this study, we chose cells from the putative reservoir host E. helvum for our analyses. However, it has to be noted that the described findings might not hold true for cell lines from other species of this large mammalian taxon. A study by Hölzer et al. has recently analyzed the IFN-induced transcriptome of Yangochiroptera bat cells in great detail, revealing substantial differences to published findings on Yinpterochiroptera bats . The authors not only describe the lack of multiple ISGs that were described in Yinpterochiroptera bats but also failed to confirm earlier findings of IFN-α gene expansion or high baseline IFN expression in bat cell lines. These discrepancies further highlight the need for bat-specific tools and an overall caution regarding broad and general statements on bat immune responses. Future studies should not only consider multiple bat species but also implement an increased repertoire of cell culture models to further elucidate the interaction of lyssaviruses and the immune system of distinct bat representatives.” 

While the authors carefully worked out promoter/concentration differences in the EiD cell lines the authors need to mention in the discussion how this is all done with artificial stimulation (transfected rna / rvfv reporter etc) and how it may differ with real infection. They should also indicate references etc showing how LBV/RBV infection would be expected to activate the same RIG-I/MDA-5 signaling pathways as VSV RNA.

There is also limited discussion on the differences between IFNs and if other (for example alpha, omega, kappa) promoters are expected to be affected in a similar fashion. There was limited discussion on ISG54 promoter whereby no species differences were observed and how what may be more relevant during infection - P's effect on IFN-b promoter compared to direct on ISG induction visa ISRE elements. (a comparison of ISG54 induction in the bat cells would be ideal).With that in mind there are other studies showing high ISG induction and characterizing IFN induction (the authors own work, ref. 56 and also https://pubmed.ncbi.nlm.nih.gov/33147460/) that are highly relevant to this study and should be discussed. Additionally there is data showing unique bat-specific IRf3 phospho sites that may be relevant to activation (and viral-induced inhibition from TBK1) that could be discussed - https://www.cell.com/iscience/fulltext/S2589-0042(20)30142-5.

Reply: We thank the reviewer for these thoughtful comments and suggestions. We implemented the suggested points in our substantially revised discussion. We comment on methods for stimulating cells and differences to virus infection. In addition, we discuss intracellular Rhabdovirus sensing and RIG-I/MDA5 involvement.

Lines 563-578: “. Of note, overexpression of signaling molecules as well as transfection of exogenous RNA are both well-established methods to analyze immune pathways and highly sensitive and fine-tuned IFN-ß promoter reporter assays. While using such stimuli enables specific targeting of certain components of immune pathways, virus infections are less artificial and elicit more complex cellular responses During replication, VSV produces different RNA species that induce both RIG-I and MDA5, although RIG-I seems to be predominantly involved in generating an efficient antiviral immune response . Comparably, RIG-I was identified as the primary sensor molecule for rabies virus as prototypic representative of the virus family Rhabdoviridae, which also encompasses VSV . In addition. rabies virus was also found to induce MDA5-dependent IFN signaling , whereas in vivo experiments suggested a superior role for RIG-I activation in survival and IFN production upon both rabies virus and VSV infection. While using exogenous RNA stimuli simplifies complex virus-host interactions and focuses on the relevant signaling pathways, detrimental effects of virus infection like transcriptional or translational shutoff are avoided . “

The discussion was further complemented by a brief comparison of additional IFN subtypes/classes and implications during lyssavirus infection or P protein involvement.

Lines 586-595: “In human cells, the induction of other IFN subtypes, such as IFN-α largely relies on the same signaling molecules as IFN-ß, including TBK-1 and IKKε, but does not require NF-κB signals for gene expression . IFN-α is not as well characterized as IFN-ß in the context of rabies virus infection and seems to play a subordinate role in antiviral signaling and was therefore not assessed in this study. Due to signaling through TBK-1 and IKKε, the major targets of RVP interference, a comparable RVP-based inhibition of IFN-α promoter activation is highly likely. Interestingly, IFN-λ has very recently been shown to attenuate rabies virus infection and induce antiviral genes, although P protein-based interference with the induction of distinct IFN classes is less well described .”

Regarding ISG54/ISRE activation in bat cells, we applied the established ISG54 reporter construct in EidLu/20.2 cells (S1 Fig D). Comparable to our results in the human cell line, LBV P showed a slightly increased capacity to block ISRE activation as compared to RVP. ISRE sequences in bats are not well described and information on ISG54-ISRE in E. helvum is not available. Despite the relatively conserved nature of ISRE motifs, species-specific differences cannot be ruled out. Nevertheless, because of a robust ISG54/ISRE induction that we observed in the E. helvum cell line and the highly specific reduction by RVP and LBV P, we are confident that the available human reporter construct is suitable to address the reviewer’s comment, hinting towards a slightly elevated STAT-interference by LBV P over RVP (please also see reply to comment 2 by reviewer 1 for details) 

Revised S1D Fig

We further discussed several of the relevant studies that were mentioned by the reviewer regarding IFN induction, ISG expression, and unique features of bat cell immunity that were recently described: 

Lines 615-625: “A study by Hölzer et al. has recently analyzed the IFN-induced transcriptome of Yangochiroptera bat cells in great detail, revealing substantial differences to published findings on Yinpterochiroptera bats . The authors not only describe the lack of multiple ISGs that were described in Yinpterochiroptera bats but also failed to confirm earlier findings of IFN-α gene expansion or high baseline IFN expression in bat cell lines. These discrepancies further highlight the need for bat-specific tools and an overall caution regarding broad and general statements on bat immune responses. Future studies should not only consider multiple bat species but also implement an increased repertoire of cell culture models to further elucidate the interaction of lyssaviruses and the immune system of distinct bat representatives.” 

Lines 640-648: “ A recent study by Irving et al. elegantly describes the influence of interferon regulatory factors 1/3/7 on ISG expression in bat cells . Besides showing high expression levels of these central antiviral regulators in bat cells and tissue, which translates to elevated ISG expression, the authors also describe a prolonged IFN-like antiviral signature and novel bat-specific antiviral mechanisms. In light of these findings, developing bat-specific tools to integrate IFN induction pathways, on the one hand, and expression of ISGs as antiviral effectors, on the other hand, might be an important step towards decoding the unique antiviral immune response of bats. “

1. Gupta AK, Blondel D, Choudhary S, Banerjee AK. The phosphoprotein of rabies virus is phosphorylated by a unique cellular protein kinase and specific isomers of protein kinase C. J Virol. 2000;74(1):91-8. Epub 1999/12/10. doi: 10.1128/jvi.74.1.91-98.2000. PubMed PMID: 10590095; PubMed Central PMCID: PMCPMC111517.

2. Mavrakis M, McCarthy AA, Roche S, Blondel D, Ruigrok RW. Structure and function of the C-terminal domain of the polymerase cofactor of rabies virus. J Mol Biol. 2004;343(4):819-31. Epub 2004/10/13. doi: 10.1016/j.jmb.2004.08.071. PubMed PMID: 15476803; PubMed Central PMCID: PMCPMC7173060.

---

## [Decision Letter · Decision Letter 1]

11 Feb 2022

Reduced IFN-ß inhibitory activity of Lagos bat virus phosphoproteins in human compared to Eidolon helvum bat cells

PONE-D-21-25746R1

Dear Dr. Müller,

We’re pleased to inform you that your manuscript has been judged scientifically suitable for publication and will be formally accepted for publication once it meets all outstanding technical requirements.

Kind regards,

Zheng Xing

Academic Editor

PLOS ONE

Additional Editor Comments (optional):

Reviewers' comments:

Reviewer's Responses to Questions

**Comments to the Author**

1. If the authors have adequately addressed your comments raised in a previous round of review and you feel that this manuscript is now acceptable for publication, you may indicate that here to bypass the “Comments to the Author” section, enter your conflict of interest statement in the “Confidential to Editor” section, and submit your "Accept" recommendation.

Reviewer #1: All comments have been addressed

2. Is the manuscript technically sound, and do the data support the conclusions?

Reviewer #1: Yes

3. Has the statistical analysis been performed appropriately and rigorously? 

Reviewer #1: N/A

4. Have the authors made all data underlying the findings in their manuscript fully available?

Reviewer #1: Yes

5. Is the manuscript presented in an intelligible fashion and written in standard English?

Reviewer #1: Yes

6. Review Comments to the Author

Reviewer #1: (No Response)

7. PLOS authors have the option to publish the peer review history of their article (what does this mean?). If published, this will include your full peer review and any attached files.

Reviewer #1: **Yes: **Karl-Klaus Conzelmann

---

## [Editor Report · Acceptance letter]

23 Feb 2022

PONE-D-21-25746R1 

Reduced IFN-ß inhibitory activity of Lagos bat virus phosphoproteins in human compared to Eidolon helvum bat cells 

Dear Dr. Müller:

I'm pleased to inform you that your manuscript has been deemed suitable for publication in PLOS ONE. Congratulations! Your manuscript is now with our production department. 

Kind regards, 

on behalf of

Dr. Zheng Xing 

Academic Editor

PLOS ONE